# A cost analysis of implementing mobile health facilitated tuberculosis contact investigation in a low-income setting

Patricia Turimumahoro[1], Austin Tucker[2], Amanda J. Gupta[1,3,4], Radhika P. Tampi[2], Diana Babirye[1], Emmanuel Ochom[1], Joseph M. Ggita[1], Irene Ayakaka[1], Hojoon Sohn[5], Achilles Katamba[1,6], David Dowdy[1,2], J. Lucian Davis[1,2,7] *

1 Uganda Tuberculosis Implementation Research Consortium, Kampala, Uganda, 2 Department of Epidemiology, Johns Hopkins Bloomberg School of Public Health, Baltimore, MD, United States of America, 3 Department of Epidemiology of Microbial Diseases, Yale School of Public Health, New Haven, CT, United States of America, 4 Johns Hopkins Bloomberg School of Public Health, Baltimore, MD, United States of America, 5 Department of Preventative Medicine, Seoul National University College of Medicine, Seoul, South Korea, 6 Clinical Epidemiology Unit, Makerere University, College of Health Sciences Kampala, Kampala, Uganda, 7 Pulmonary, Critical Care and Sleep Medicine Section, Yale School of Medicine, New Haven, CT, United States of America

* lucian.davis@yale.edu

**Data Availability Statement:** Data are held in the Dryad online repository, available at https://datadryad.org/stash/share/02ECnaLd-vQstsxpf3_L8tWynB6DG8ILjKcAHqqdK3g.

## Abstract

### Introduction

Mobile health (mHealth) applications may improve timely access to health services and improve patient-provider communication, but the upfront costs of implementation may be prohibitive, especially in resource-limited settings.

### Methods

We measured the costs of developing and implementing an mHealth-facilitated, home-based strategy for tuberculosis (TB) contact investigation in Kampala, Uganda, between February 2014 and July 2017. We compared routine implementation involving community health workers (CHWs) screening and referring household contacts to clinics for TB evaluation to home-based HIV testing and sputum collection and transport with test results delivered by automated short messaging services (SMS). We carried out key informant interviews with CHWs and asked them to complete time-and-motion surveys. We estimated program costs from the perspective of the Ugandan health system, using top-down and bottom-up (components-based) approaches. We estimated total costs per contact investigated and per TB-positive contact identified in 2018 US dollars, one and five years after program implementation.

### Results

The total top-down cost was $472,327, including $358,504 (76%) for program development and $108,584 (24%) for program implementation. This corresponded to $320-$348 per household contact investigated and $8,873-$9,652 per contact diagnosed with active TB

**Funding:** This study was supported by National Institutes of Health through grants awarded to JLD (R01AI104824 and D43TW009607). The funder had no role in study design, data collection and analysis, decision to publish, or preparation of the manuscript.

**Competing interests:** The authors have declared that no competing interests exist.

over a 5-year period. CHW time was spent primarily evaluating household contacts who returned to the clinic for evaluation (median 30 minutes per contact investigated, interquartile range [IQR]: 30–70), collecting sputum samples (median 29 minutes, IQR: 25–30) and offering HIV testing services (median 28 minutes, IQR: 17–43). Cost estimates were sensitive to infrastructural capacity needs, program reach, and the epidemiological yield of contact investigation.

## Conclusion

Over 75% of all costs of the mHealth-facilitated TB contact investigation strategy were dedicated to establishing mHealth infrastructure and capacity. Implementing the mHealth strategy at scale and maintaining it over a longer time horizon could help decrease development costs as a proportion of total costs.

## Introduction

Tuberculosis (TB) is among the leading causes of death due to an infectious disease worldwide, with approximately 7 million new TB cases diagnosed in 2020 [1]. Low TB case detection rates represent a major gap in the TB care cascade in high burden countries, and more than 30% of estimated incident TB cases continue to go undiagnosed and/or unreported [1–3]. This problem has impeded progress toward the global End TB targets [4]. Patient-centered interventions that can facilitate early case detection and reduce barriers to TB care are thus an important public health and global health priority.

Contacts of TB patients have a substantial risk of developing active TB within the first one to two years after exposure [5]. and in high-income settings, interventions focusing on contacts of index TB patients (*e.g.*, contact investigation) have become an important priority for TB control and elimination [6]. TB contact investigation involves teams of health care workers visiting households or workplaces of people diagnosed with active TB to identify and refer those with TB-specific symptoms or key risk factors for further clinical and bacteriologic TB evaluation. Recent evidence suggests that contact investigation has the potential to improve TB case detection in high prevalence settings relative to passive case-detection services in health facilities [7]. However, barriers to acceptance and completion, operational complexities, and resource constraints have limited wide adoption of contact investigation in low-and middle-income countries[2,8,9].

To address these challenges, we developed a home-based, mHealth-facilitated household contact investigation strategy and evaluated it in a pragmatic, prospective, household randomized trial [10]. Compared to routine contact investigation delivered by community health workers (CHWs), the mHealth-facilitated contact investigation intervention included home-based HIV testing and TB evaluation, collection and transport of sputum samples, and follow-up communications using automated short messaging services (SMS). The strategy was feasible and acceptable but not more effective than routine contact investigation because of implementation challenges [11,12]. Nonetheless, another area of uncertainty in the mHealth field is the limited and heterogeneous evidence on the costs and cost effectiveness of mHealth strategies [13] including some evidence of high up-front costs [14], which may in turn act as a barrier to ongoing research and innovation. Therefore, to characterize the resource implications of mobile health interventions more fully, we conducted a comprehensive assessment of the

costs of development, implementation, and maintenance of home-based, mHealth-facilitated TB contact investigation in Kampala, Uganda.

## Materials and methods

### Study design and setting

We estimated the costs of developing, implementing, and maintaining a home-based, mHealth-facilitated household TB contact investigation intervention in Kampala, Uganda, from the health system perspective, using both a "top-down" and "bottom-up" (components-based) approach. In this setting, TB contact investigation involved CHWs visiting the homes of TB patients, screening all contacts for TB symptoms, and recording their findings using a customized electronic survey application (CommCare, Dimagi, Boston, USA). The application employed decision-support logic to identify contacts requiring evaluation for TB and prompted CHWs to collect a sputum sample and offer HIV testing to eligible household members. The application also delivered personalized, automated text messages to each participant providing follow-up instructions, clinic visit reminders, and TB test results. In the routine care arm, automated text messages were not sent, and all contacts needing TB evaluation were referred to the clinic. The home-based strategy sought to increase the proportion of contacts fully evaluated for TB by reducing the need for contacts to travel to clinics.

To comprehensively evaluate the costs of the mHealth-facilitated intervention, we divided the program into two phases and evaluated the costs accrued in each phase. The development phase, which lasted 30 months (February 2014-July 2016), consisted of formative research, software customization, and pilot testing. Activities during this phase included, but were not limited to, development of decision-support logic, integration of fingerprint identification technology and automated short messaging service (SMS) technology; pilot testing; and optimization of technological components. This was followed by the implementation phase, which occurred over 12 months (July 2016 –July 2017) in the context of a cluster-randomized trial. A total of 919 contacts were randomized at the household level, including 471 contacts in the intervention arm who are the focus of this cost analysis. The trial observed a marginal probability of completing TB evaluation of 14% (95% CI 8–20) in intervention households and 15% (95% CI 9–21) in routine care households, representing a difference of -1% (95% CI -9% to 7%, p = 0.81) [10]. **Fig 1** shows a conceptual outline of project phases, activities, and timelines.

### Estimation of program development costs

To estimate the cost of developing, implementing, and maintaining the mHealth-facilitated contact investigation intervention, we retrospectively collected programmatic costs of each phase of development from the health system perspective using health facility and study budget estimates. All cost and volume estimates extracted from study and health center financial records, and key informant interviews were performed with study staff and health facility administrators to confirm which budgetary items mapped to specific expenditures for program development. Cost components were appropriately mapped to specific thematic expenditure categories: human resource costs, capital costs, recurrent costs, overhead costs, and building space costs. Human resource costs included salaries of a coordinator, data manager, laboratory manager, IT officer and CHWs. Capital costs included investment in hardware and software for mHealth, a vehicle, and cost to adapt the intervention to the local setting. Recurrent costs included expenditure on consumables such as laboratory supplies, internet, and text messages. Overhead costs included operational costs such as those for supervision teams and patient care at the clinic. Building space included the space occupied by the supervision teams and patient rooms. Since budget allocations were similar across facilities with similar clinical

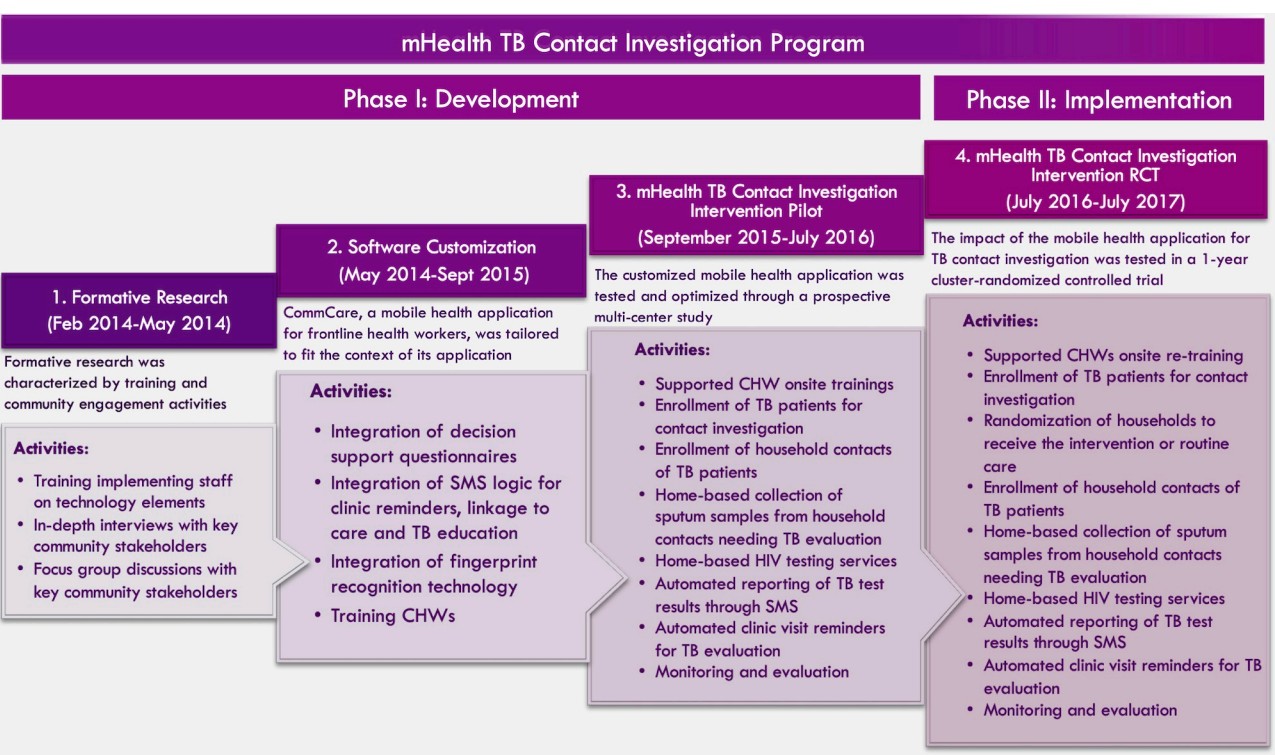

**Fig 1. A detailed description of program activities in the different phases of implementation.**

capacity, patient visit volumes, and scope of service delivery, we compiled cost data from four of the seven participating health centers and extrapolated these costs for the remaining three health facilities. Full details on cost components, according to study phase, are summarized in the Supporting Information, **S1 Table**.

## Estimation of program implementation costs

We assessed the cost of program implementation using both a top-down and bottom-up approach. Top-down costing of program implementation was performed similarly to the top-down estimation of program development costs described above. Bottom-up costing was performed using an activity-based approach. Specifically, we conducted a time-and-motion (TAM) study on consecutive days between March and August 2017 asking CHWs to record start and end times for each discrete contact investigation activity, as defined in the Supporting Information, **S2 Table**. Clinic-based activities included: TB index patient recruitment, waiting time, contact evaluation, and other activities. Community-based activities included: travel to and from the household, TB education and counseling, household contact screening for TB symptoms, HIV testing, sputum collection, HIV testing, and confirmation of phone contacts for SMS messaging. Using the top-down approach, we summed unit costs in five primary categories: human resources, capital investments, building space, overhead costs, and recurrent costs. Human resource costs were estimated by enumerating staffing levels for the entire development phase and multiplying monthly staff salaries by the percent effort contributed to program development (50%), based on opinions from key informants. Capital investments included software, hardware, and formative work. The cost of building space utilized for patient services was approximated as 5% of the cost of the entire building and operational

costs for the program as 6–7% of clinic operational costs. The cost estimate for one square meter of building space was based on local rates suggested by clinic administrators and multiplied by the measured area of the building. Recurrent costs accrued during the development of the program were summarized as software, lab consumables, and program evaluation costs for routine quality assurance. The total cost of program development was then calculated as the total of all five categories over the entirety of the development phase. The cost of program implementation was calculated in a similar fashion, and total program costs were estimated as the sum of program development plus program implementation costs.

## Total program costs: Bottom-up

For bottom-up costing, program development costs were assessed as annual costs based on corresponding estimates of useful life years for each component (between 5 and 30 years, based on key informant interviews). These were converted to an estimated cost per minute-use, based on the estimated total number of operational minutes per year: 8 hours per day, 5 days per week, 46 weeks per year). The bottom-up costs of program implementation were estimated by multiplying the median time estimates to perform each activity (estimated using TAM studies as above) by the cost per minute for each resource type. Resource use was categorized as 1) direct human resources; 2) capital equipment; 3) program overhead (operational costs and recurrent costs); and 4) building space costs. We summed the unit costs for each category to calculate the total activity-based cost per household contact investigated. Some activities (*e.g.*, TB index patient recruitment) were associated with a specific index case, not a specific household contact. For these activities, we divided these unit time estimates by the average number of household contacts observed per index patient to get the time estimate for each activity per household contact investigated.

## Analysis

We assumed five useful life years for all program development costs except for building space and vehicles, for which we assumed an expected useful life of 30 years. All capital costs were depreciated linearly using a 3% discount rate. We divided the total program costs by the total number of patients enrolled and the total number of new TB diagnoses made to estimate the cost per household contact investigated and the cost per TB diagnosis made. All costs are reported in 2018 US dollars. All costs measured in Ugandan Shillings were updated to 2018 using the Ugandan GDP deflator [15] and converted to US dollars at the average annual exchange rate for 2018 [16].

## Sensitivity analysis

This program required a large initial investment in technology development, infrastructure, and equipment with a low marginal cost for including additional patients during program implementation. As such, we performed a sensitivity analysis to estimate costs under "continued implementation", under the assumption that the one-year intervention could be continued for four additional years at the same volume without any additional development costs (*i.e.*, continued implementation costs only). To perform this sensitivity analysis, we used top-down cost estimates and allocated costs incurred at the program level, clinic level, and contact level. We then multiplied costs at the clinic level by the number of clinics included in the program and costs at the contact level by the number of contacts screened per clinic. To explore differences in clinic capacity over different years, we performed a three-way sensitivity analysis in which we simultaneously varied the numbers of clinics that might be covered by the mHealth program (in increments of 10, from 10 to 50), the mean annual contacts evaluated per clinic (in

increments of 50, from 100 to 350), and the average annual contact positivity rate (at four levels, from 0.025 to 0.042). The maximum number of clinics that could be covered was based on the expert opinion of research staff and implementers, and the annual number of contacts who would be screened per clinic was estimated using the observed number of patients per facility (plus or minus 2 standard deviations from the mean as minimum and maximum values).

## Ethical considerations

The School of Medicine Research Ethics Committee at Makerere University; the Uganda National Council for Science and Technology; and the Yale University Human Investigation Committee approved the study protocol, informed consent forms, and assent forms.

## Results

In the 12 months of program implementation, 190 index TB patients with 471 household contacts were randomized to receive the intervention. The TB case notification rate at the seven contributing health centers ranged from 15 to 67 TB cases per month. Of the 471 contacts, 106 (23%) had TB symptoms, ranging from 14% to 33% across health centers (**Table 1**).

Using a top-down approach, the total cost of the mHealth TB contact investigation intervention was estimated at $472,327, of which program development accounted for $358,504 (76%) and program implementation for $113,823 (24%). Human resource costs accounted for

**Table 1. Clinic characteristics.**

| Clinic Name* | Total | Naguru | Kawaala | Kisenyi | Kisugu | Kiswa | Kitebi | Komamboga |
|---|---|---|---|---|---|---|---|---|
| | | Hospital | HC III | HC IV | HC III | HC III | HC III | HC III |
| Location | | Urban | Urban | Urban | Urban | Urban | Rural | Rural |
| **Service Statistics** | | | | | | | | |
| Average monthly TB case notifications** | | 32 | 39 | 67 | 15 | 15 | 16 | 16 |
| Total households enrolled*** | 163 | 37 | 27 | 29 | 10 | 30 | 11 | 19 |
| Average monthly household visits made | 13.6 | 3.1 | 2.3 | 2.4 | 0.83 | 2.5 | 0.92 | 1.6 |
| Total contacts enrolled | 471 | 117 | 76 | 97 | 21 | 91 | 24 | 45 |
| Average contacts per household | 2.9 | 3.2 | 2.8 | 3.3 | 2.1 | 3.0 | 2.2 | 2.4 |
| Total symptomatic TB household contacts | 106 | 25 | 13 | 32 | 4 | 13 | 5 | 14 |
| Proportion of contacts needing TB evaluation | 22.5% | 21% | 17% | 33% | 19% | 14% | 21% | 31% |
| **Staffing**** | | | | | | | | |
| Doctors/Clinical officer | 1 | 1 | 0 | 0 | 0 | 0 | 0 | 0 |
| Nurses | 14 | 2 | 2 | 2 | 2 | 2 | 2 | 2 |
| CHWs | 14 | 1 | 3 | 3 | 1 | 2 | 2 | 2 |
| Total | | 4 | 5 | 5 | 3 | 4 | 4 | 4 |
| **Monthly Workload/Staffing Ratio** | | | | | | | | |
| Average monthly contacts screened***** | 39 | 9.8 | 6.3 | 8.1 | 1.8 | 7.6 | 2.0 | 3.8 |
| CHW-to-Contact Ratio | 1:34 | 1:117 | 1:25 | 1:32 | 1:21 | 1:46 | 1:12 | 1:23 |

**Abbreviations:** CHW, Community health worker; HC, Health Centre; TB, tuberculosis.

**Legend:** * Health care delivery is through a decentralized framework consisting of Village Health Teams, Health Centre (HC) II, Health Centre III, Health Centre IV/ Referral Hospital, Regional Referral Hospital and a National Referral Hospital.

** The average monthly TB case notification rate was calculated based on TB case notifications between January 2017 and December 2017.

*** The households enrolled between July 2016 and July 2017 that were eligible to enroll into the study and were randomized to receive the intervention.

**** These staffing characteristics and numbers include only those working in the TB specialty units in these facilities.

*****The average monthly household contacts screened was calculated by dividing the total contacts screened by the period of implementation (12 months).

$178,542 (38%), capital assets for $156,091 (33%), and recurrent costs for $74,965 (16%), overhead costs for $51,202, and building space for $11,525 (2%) (**Table 2A**).

The mHealth TB contact investigation intervention was estimated to cost $1,003 per household contact investigated or $27,748 per positive contact found. Assuming that the intervention was continued for an additional four years at similar capacity, the program was estimated to cost $348 per household contact investigated, or $9,652 per positive contact diagnosed (**Table 2B**).

In the time and motion survey, we observed a total of 12,100 person-minutes from 11 discrete activities across seven trial clinics. A total of 5,496 (45%) person-minutes of clinic-

**Table 2.** a. Granular unit cost estimates for each phase of implementation of an mHealth-facilitated TB contact investigation program. b. Top-down cost estimates for household contact investigation of tuberculosis in Uganda.

**A.**

| Resource Category* | Development Phase | | | | Implementation Phase | |
|---|---|---|---|---|---|---|
| | Formative Research | Software Customization | Program Pilot | Development Costs | Implementation Costs | Total Costs |
| **Human resource costs** | **$ 14,979** | **$ 61,369** | **$ 49,487** | **$ 125,835 (35%)** | **$ 52,707 (46%)** | **$178,542 (38%)** |
| Community health workers | - | - | $ 8,714 | $ 8,714 | $ 9,506 | $ 18,220 |
| Administrative staff | $ 9,400 | $ 39,950 | $ 25,850 | $ 75,200 | $ 28,200 | $ 103,400 |
| Software consultants | $ 5,579 | $ 21,419 | $ 14,923 | $ 41,921 | $ 15,001 | $ 56,922 |
| **Capital costs** | **$ 100,200** | **$ 28,855** | **$ 24,656** | **$ 153,710 (43%)** | **$ 2,381 (2%)** | **$ 156,091 (33%)** |
| Software | $ 55,792 | - | - | $ 55,792 | - | $ 55,792 |
| Hardware | - | $ 18,277 | $ 22,410 | $ 40,687 | - | $ 40,741 |
| Training | $ 25,104 | $ 10,577 | $ 2,246 | $ 37,927 | $ 2,381 | $ 40,309 |
| Vehicle | $ 16,738 | - | - | $ 16,738 | - | $ 16,738 |
| Adaptation to local setting | $ 2,566 | - | - | $ 2,566 | - | $ 2,566 |
| **Recurrent costs** | - | **$ 20,390** | **$ 21,517** | **$ 41,907 (12%)** | **$ 27,819 (24%)** | **$ 74,965 (16%)** |
| Software hosting plan | - | $ 20,390 | $ 12,536 | $ 32,926 | $ 13,115 | $ 46,040 |
| Supplies | - | - | $ 8,927 | $ 8,927 | $ 14,705 | $ 23,631 |
| Program evaluation | - | - | - | - | $ 5,239 | $ 5,239 |
| SMS service | - | - | **$ 54** | **$ 54** | - | - |
| **Overhead costs** | **$ 4,646** | **$ 16,636** | **$ 15,770** | **$ 37,051 (10%)** | **$ 14,150 (12%)** | **$ 51,202 (11%)** |
| Overhead (Patient care) | - | - | $ 1,132 | $ 1,132 | $ 1,235 | $ 2,367 |
| Overhead (Administrative) | $ 4,646 | $ 16,636 | $ 14,638 | $ 35,919 | $ 12,915 | $ 84,754 |
| **Building space** | - | - | - | - | **$ 11,525 (10%)** | **$ 11,525 (2%)** |
| Patient care | - | - | - | - | $ 11,525 | $ 11,525 |
| **Total (Row Proportion)** | **$ 119,825 (25%)** | **$ 127,249 (27%)** | **$ 111,430 (24%)** | **$ 358,504 (76%)** | **$ 113,823 (24%)** | **$ 472,327** |

**B.**

| Resource Category | Cost per contact investigated | | | Cost per TB positive contact found | |
|---|---|---|---|---|---|
| | | n = 471 | | | n = 17 |
| | Total Program Costs | Program Only Costs* | Continued Implementation Costs** | Program Only Costs* | Continued Implementation Costs** |
| Human resource costs | $178,542 (38%) | $379 | $150 | $10,502 | $4,143 |
| Capital costs | $156,091 (33%) | $331 | $72 | $9,182 | $1,999 |
| Recurrent costs | $74,965 (16%) | $159 | $78 | $4,407 | $2,170 |
| Overhead costs | $51,202 (11%) | $109 | $47 | $3,012 | $1,304 |
| Building space | $11,525 (2%) | $24 | $1 | $678 | $35 |
| **Total** | **$472,327** | **$1,003** | **$348** | **$27,784** | **$9,652** |

**Abbreviations:** SMS, short-messaging service.

**Table 2a Legend:** * Costs in 2018 US dollars; percentages displayed are column percentages, unless otherwise specified.

**Table 2b Legend:** *Program only costs assume that all intervention activities stop at the end of the observed 12-month program implementation period.

**Continued implementation costs assume continued implementation of the program for a total of five years (at a similar annual volume as observed in the first year).

based activities and 6,604 (55%) person-minutes of community-based activities were observed. In the clinic, CHWs spent approximately 1.2 hours per household contact investigated, with the most time spent evaluating contacts returning to the clinic (median 30 person-minutes per contact evaluated enrolled, interquartile range [IQR]: 30–70). In the community, CHWs spent approximately 3.5 hours per contact investigated, with sputum collection (median 29 person-minutes per contact investigated, IQR 25–30) and offering HIV testing services (median 28 minutes per contacts investigated, IQR: 17–43) found to be the most time-consuming activities (Supporting Information, S3 Table). After enumerating all costs and activity times, our bottom-up cost estimate of program costs was $320 per household contact investigated or $8,873 per TB positive contact diagnosed, with component cost estimates as summarized in Table 3.

Program capacity (both the number of participating facilities and the number of household contacts investigated per facility) had a large effect on the estimated cost per positive contact diagnosed (Fig 2). At the observed average annual contact positivity rate (0.036), over half of the capacity scenarios projected a cost per positive contact diagnosed of less than $600 –substantially lower than the estimates of $8,873 (bottom-up) and $9,652 (top-down) at the observed capacity of 7 clinics and 67 contacts screened per facility. Under the highest capacity projected (50 facilities with an average of 300 contacts investigated per facility per year), and the average annual contact positivity rate (0.036), we estimated that this program could be developed and implemented at a total cost of $459 per contact diagnosed with TB. Higher TB prevalence among contacts resulted in even lower costs per person diagnosed.

**Table 3. Bottom-up unit cost estimates for each activity and resource category per household TB contact investigated.**

| Activity category | Activity type | Total person minutes per contact (%) | Median person minutes per contact (IQR) | Human resources | | Capital costs | | Recurrent costs | | | Overhead costs | Building space | Total cost per contact investigated |
|---|---|---|---|---|---|---|---|---|---|---|---|---|---|
| | | | | CHWs | Program staff & mHealth team | IT | Other | SMS | Software | Supplies | | | |
| *Clinic activities* | TB patient recruitment | 1,002 (8%) | 12 (5–26) | $0.06 | $7.45 | $2.27 | $1.42 | $0.01 | $2.20 | $2.38 | $2.41 | $0.47 | $18.68 |
| | Waiting for clients | 3,565 (29%) | 21 (10–42) | $0.11 | $13.04 | $3.97 | $2.49 | $0.01 | $3.85 | $4.16 | $4.22 | $0.83 | $32.68 |
| | Contact evaluation | 419 (3%) | 30 (30–70) | $0.16 | $18.63 | $5.67 | $3.56 | $0.01 | $5.50 | $5.94 | $6.02 | $1.18 | $46.68 |
| | Other | 510 (4%) | 10 (5–21) | $0.05 | $6.21 | $1.89 | $1.19 | $0.00 | $1.83 | $1.98 | $2.01 | $0.39 | $15.56 |
| *Community activities* | Travel | 1,707 (14%) | 21 (14–32) | $0.11 | $13.04 | $3.97 | $2.49 | $0.01 | $3.85 | $4.16 | $4.22 | $0.83 | $32.68 |
| | TB education & counselling | 335 (3%) | 7 (5–16) | $0.04 | $4.35 | $1.32 | $0.83 | $0.01 | $1.28 | $1.39 | $1.41 | $0.28 | $10.89 |
| | Contact screening | 2,104 (17%) | 20 (11–38) | $0.11 | $12.42 | $3.78 | $2.37 | $0.01 | $3.67 | $3.96 | $4.02 | $0.79 | $31.13 |
| | HIV testing | 1,089 (9%) | 28 (17–43) | $0.15 | $17.39 | $5.30 | $3.32 | $0.01 | $5.13 | $3.71 | $5.62 | $1.11 | $41.74 |
| | Sputum collection & HIV testing | 1,063 (9%) | 19 (5–30) | $0.10 | $11.80 | $3.59 | $2.26 | $0.01 | $3.48 | $4.29 | $3.82 | $0.75 | $30.09 |
| | Sputum collection | 258 (2%) | 29 (25–30) | $0.15 | $18.01 | $5.49 | $3.44 | $0.01 | $5.32 | $0.57 | $5.82 | $1.14 | $39.96 |
| | Phone number confirmation | 50 (0%) | 13 (9–16) | $0.07 | $8.07 | $2.46 | $1.54 | $0.01 | $2.38 | $2.58 | $2.61 | $0.51 | $20.23 |
| **Total cost per contact investigated, by cost category** | | | | $1.10 | $130.40 | $39.72 | $24.93 | $0.10 | $38.50 | $35.12 | $42.17 | $8.29 | $320.35 |

**Abbreviations:** IQR, Interquartile range; TB, tuberculosis; CHWs, community health workers; SMS, short messaging service.

## Discussion

In resource-limited settings, mHealth technologies are being widely implemented, not only for TB contact investigation [17–20], but also for a wide array of other health-related interventions. More broadly, eHealth is being rapidly adopted worldwide and has been endorsed by the WHO [18]. In this economic evaluation we provide context on the adaptation of mHealth for a home-based TB contact investigation intervention in a low-income setting. Specifically, we found that 76% of the total cost of the program was incurred during development, before the recruitment of a single participant. The total program cost was therefore estimated at $320-$348 per contact investigated and $8,873-$9,652 per new diagnosis of TB–a high cost relative to many other programs [13,21,22]. Importantly, these costs could be reduced to under $600 per new TB diagnosis simply by expanding capacity (extending from one to five years, increasing the number of clinics participating, and optimizing the volume of household contacts evaluated in each clinic). These findings illustrate that, when implementing mHealth and other interventions with substantial development costs in resource-limited settings, the feasible scope and duration of the program, as well as the expected yield, must be considered to evaluate whether a meaningful return on investment is likely.

Assuming that the maximum capacity of participants could be achieved, our projected costs of mHealth-facilitated contact investigation are comparable to the estimated costs of contact investigation (without mHealth) in other low-income settings, with one economic evaluation of contact investigation in Uganda estimating a cost of US$878 per new TB case identified [23,24]. This finding suggests that mHealth implementation may be economically viable in this context if sufficient patient volumes can be achieved. The drivers of cost in the development phase of this project included capital investments in technology (42%) and human resources–including highly trained IT staff and implementers (36%). During the implementation phase, human resource costs for supervision became more prominent (46%), reflecting the relatively small number of participants engaged relative to high-level supervisory staff. Again, this finding highlights the need to achieve economies of scale to make such programs more cost-effective.

More broadly, the cost implications of digital health interventions in low- and middle-income countries vary depending on their scope and purpose. For example, a digital adherence program in Brazil applying two-way SMS cited a cost of $65 ($53–$105) per person enrolled [25], while an mHealth support intervention for HIV in Uganda cited an annual cost of $2.35 per patient enrolled [22]. A study in India on an mHealth-facilitated intervention to improve CHW counseling skills in maternal and newborn health suggested that start-up costs represented only 9% of total costs; this program was able to operate at a cost of $20 per woman registered [21]. Taken as a whole, these findings illustrate the heterogeneous economic implications of mHealth-facilitated health interventions designed for resource-limited settings, as well as the importance of considering what is feasible in terms of program scope and the costs of program development before embarking on large investments in mHealth infrastructure and capacity.

Our analysis adheres to transparency recommendations on scope and accuracy, and captures unit cost estimates of all the key program components [26]. While we do not report granular expenditures (*e.g.*, number of sputum cups purchased), all macro cost calculations were derived from itemized unit costs and quantities. Furthermore, our method of micro-cost estimation also provides the recommended highest level of accuracy, estimating the cost of each discrete implementation activity based on actual consumption. The observed breakdown of costs for program development versus implementation may serve as a reference for future cost-effectiveness analyses [27], and help potential implementers concisely project the budgetary implications of such a program over time. Our cost estimates using a top-down and

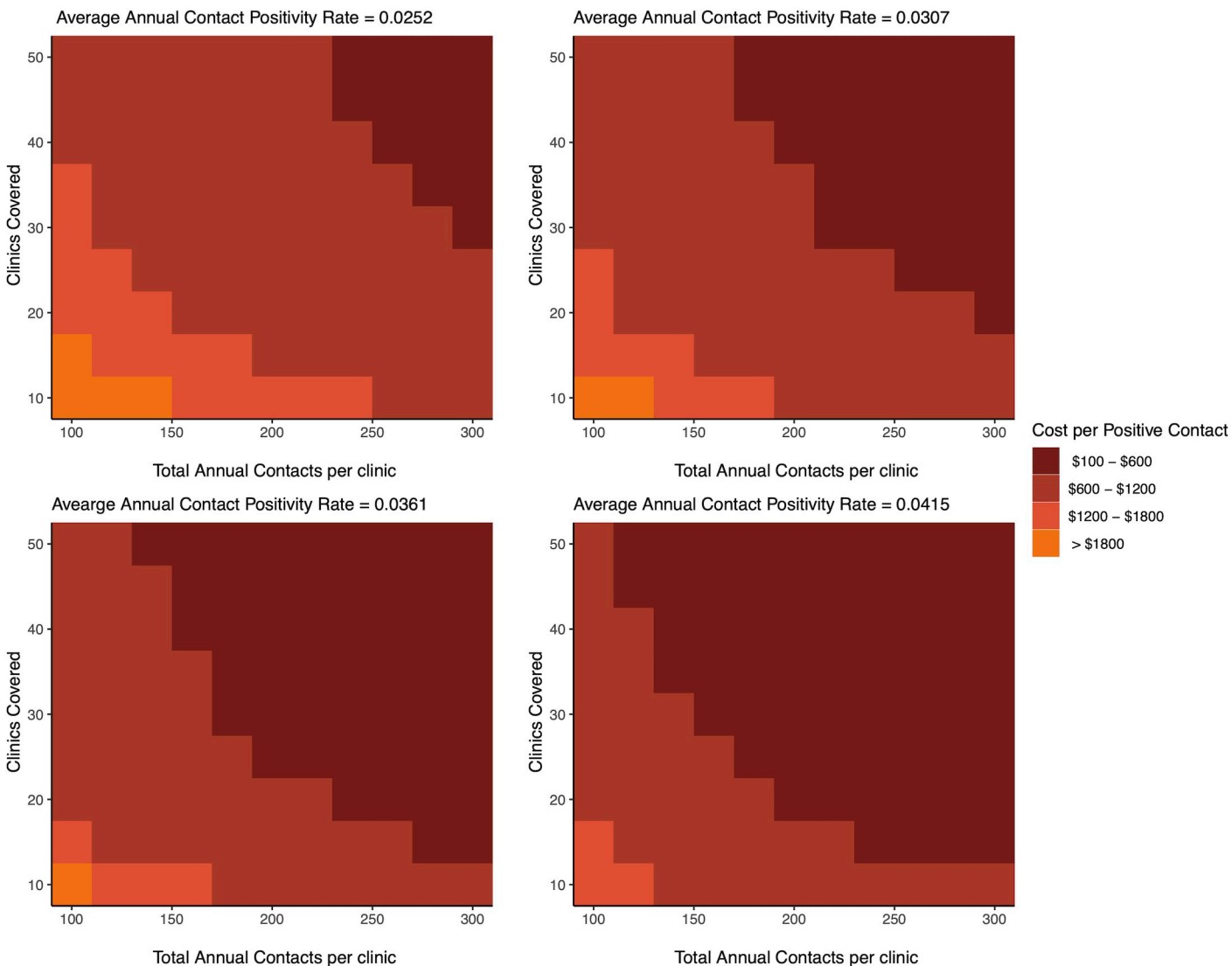

**Fig 2. Three-way sensitivity analysis of the cost per TB case detected.** Each graph represents the cost of an mHealth-facilitated TB contact investigation program as the average annual contact positivity rate (number of contacts diagnosed with TB divided by the number of contacts evaluated) varies from 0.0252 to 0.0307 to 0.0361 to 0.0415 among household contacts of TB patients. The y-axis represents the number of clinics covered by the program and the x-axis the total annual contacts at each clinic. The orange hue gradient represents the cost gradient associated with different program coverage capacities, with the darkest shade representing scenarios with the lowest programmatic costs and the lightest shade representing scenarios with the highest programmatic costs. Abbreviations: CHW, Community health worker; RCT, randomized-controlled trial; SMS, short-messaging service; TB, tuberculosis.

bottom-up approach were reasonably similar, suggesting that our estimates are robust to the precise method of costing employed [28]. The inputs for this analysis were derived from real-world implementation; while the trial did not find a significant improvement in TB case detection [10], it identified important process-related challenges that, if addressed, could improve program adoption, implementation, and maintenance [2,11,29]. A follow-up trial using human-centered design and communities of practice to overcome these barriers is currently underway [30,31], and the cost implications of the revised design and implementation processes will also be important to consider.

This study had several limitations. First, volume-based costs were collected retrospectively from budget estimates and financial records. This may have led to underestimation of the

absolute unit cost estimates of mHealth TB contact investigation due to missing records or poor recall by implementing staff. This retrospective data collection may also have compromised the delineation between programmatic and research-related costs. Second, clinic estimates of overhead costs were not readily available because of variations and inconsistency of funding at the clinic, with many of the inputs based on verbal estimates by clinic administrators. These limitations could have led to overestimation or underestimation of the cost of overheads due to recall bias. Third, we conducted a self-reported time-and-motion study by CHWs; these results are therefore subject to potential social desirability bias. Self-reported time-and-motion forms were collected daily, and data quality checks were done immediately to minimize this bias, but future evaluations could attempt to use more automated approaches to track activity times. Fourth, as a focused estimate of the cost of contact investigation, broader costs (such as patients' cost of accessing TB diagnostic and treatment services if found to be positive for TB or recommended for TB preventive therapy) that are of relevance to society were not included. Finally, scenarios for sensitivity analysis were constructed based on expert opinion and may not fully represent the potential scope of the program if scaled up more broadly.

## Conclusions

In summary, this investigation of the costs of mHealth-facilitated contact investigation for TB revealed high up-front costs for design and development of the mHealth infrastructure and capacity; these costs amounted to three-fourths of total program costs after one year of program implementation. In deciding whether to invest in similar interventions in Uganda and other resource-limited settings, careful consideration must be made as to the feasible duration and scope of program implementation to ensure a favorable return on investment.

## Supporting information

**S1 Table. Detailed list of cost components considered for the mHealth-facilitated contact investigation program.**
(DOCX)

**S2 Table. Community health worker activities identified during time-and-motion surveys.**
(DOCX)

**S3 Table. Distribution of community health worker activity time by health center per household TB contact investigated.**
(DOCX)

## Acknowledgments

The authors thank all of the community health workers on the mHealth Contact Investigation Study; AIDS Information Centre staff; all administrative, TB unit, and laboratory staff at the participating Kampala City Council Authority clinics; staff at the Department of Health at the Kampala City Council Authority; National TB and Leprosy Program staff at the Uganda Ministry of Health, and all participants of the mHealth Contact Investigation Study.

## Author Contributions

**Conceptualization:** Patricia Turimumahoro, Radhika P. Tampi, Irene Ayakaka, Hojoon Sohn, Achilles Katamba, David Dowdy, J. Lucian Davis.

**Data curation:** Diana Babirye, Emmanuel Ochom, Joseph M. Ggita, Irene Ayakaka.

**Formal analysis:** Patricia Turimumahoro, Austin Tucker, Hojoon Sohn, David Dowdy.

**Funding acquisition:** J. Lucian Davis.

**Investigation:** Radhika P. Tampi, Emmanuel Ochom, Joseph M. Ggita, J. Lucian Davis.

**Methodology:** Patricia Turimumahoro, Radhika P. Tampi, Hojoon Sohn, Achilles Katamba, David Dowdy, J. Lucian Davis.

**Project administration:** Patricia Turimumahoro, Amanda J. Gupta, Hojoon Sohn.

**Software:** Diana Babirye.

**Supervision:** Amanda J. Gupta, Irene Ayakaka, Hojoon Sohn, Achilles Katamba, J. Lucian Davis.

**Validation:** Hojoon Sohn.

**Writing – original draft:** Patricia Turimumahoro, Austin Tucker, Amanda J. Gupta, Hojoon Sohn, David Dowdy.

**Writing – review & editing:** Patricia Turimumahoro, Austin Tucker, Amanda J. Gupta, Radhika P. Tampi, Diana Babirye, Emmanuel Ochom, Joseph M. Ggita, Irene Ayakaka, Hojoon Sohn, Achilles Katamba, David Dowdy, J. Lucian Davis.

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
