## [Decision Letter · Decision Letter 0]

2 Nov 2021

PONE-D-21-01299A cost analysis of implementing mobile-health facilitated tuberculosis contact investigation in a low income settingPLOS ONE

Dear Dr. Davis,

Thank you for submitting your manuscript to PLOS ONE. After careful consideration, we feel that it has merit but does not fully meet PLOS ONE’s publication criteria as it currently stands. Therefore, we invite you to submit a revised version of the manuscript that addresses the points raised during the review process. The reviewers and editors comments are below.

We look forward to receiving your revised manuscript.

Kind regards,

Limakatso Lebina, MBChB

Academic Editor

PLOS ONE

Additional Editor Comments (if provided):

We apologies for delays in providing feedback. It was not easy to find appropriate reviewers and those agreed to review requested additional time.

Please also address the following to improve clarity and flow of the manuscript

1. Provide a clear description of the program, and indicate which activities were done in the clinics, and which ones were part of the community initiative and why.

2. The cost calculations are said to be done from the health system perspective, yet waiting times and travel costs for patients have been included. Please separate the health system costs and the patient costs.

3. Capital costs for the overall TB treatment have also been included, however this was an add on program, TB services costs were already incurred, it would be more important to the policy makers to understand the additional costs that will be required to include the mHealth component.

4. There is no information on the messages that were sent and the associated costs of sending the messages.

5. It appears that the research related costs such as patient enrolment and program evaluation have been included in the costing, as indicated above, policy makers would be more interested in the additional costs of implementing the program

Journal Requirements:

5. Please upload a new copy of Figure 2 as the detail is not clear. Please follow the link for more information: " ext-link-type="uri" xlink:type="simple">https://blogs.plos.org/plos/2019/06/looking-good-tips-for-creating-your-plos-figures-graphics/"
https://blogs.plos.org/plos/2019/06/looking-good-tips-for-creating-your-plos-figures-graphics/. 

Reviewers' comments:

Reviewer's Responses to Questions

**Comments to the Author**

1. Is the manuscript technically sound, and do the data support the conclusions?

Reviewer #1: Yes

Reviewer #2: Yes

2. Has the statistical analysis been performed appropriately and rigorously? 

Reviewer #1: Yes

Reviewer #2: Yes

3. Have the authors made all data underlying the findings in their manuscript fully available?

Reviewer #1: Yes

Reviewer #2: Yes

4. Is the manuscript presented in an intelligible fashion and written in standard English?

Reviewer #1: Yes

Reviewer #2: No

5. Review Comments to the Author

Reviewer #1: The manuscript is well-written and methodologically sound. For further improvement the authors can check the manuscript against the transparency standards for cost estimates from the reference: Fukuda H, Imanaka Y: Assessment of transparency of cost estimates in economic evaluations of patient safety programmes. J Eval Clin Pract. 2009, 15: 451-459. 10.1111/j.1365-2753.2008.01033.x.

Reviewer #2: Specific feedback to the Author:

Abstract

Introduction: Mobile health (mHealth) applications may improve timely access to health services and improve patient-provider communication but may be costly to implement, especially in resource-limited settings. This statement does not quite reflect the rationale for the study. Suggest changing to reflect that development/ upfront investment in m- interventions maybe resource intensive.

- The authors need to provide a clear aim/ research question in their abstract that aligns with the title and accurately represents the intervention. In the abstract, the authors state: We assessed the cost of development and execution of an mHealth-based program for contact investigation of tuberculosis (TB) in Kampala, Uganda however, this program has a m-health component alongside face to face intervention (i.e., community Health Worker (CHW)-led home-based specimen collection) therefore the authors should continue to refer to this program as a mobile-health facilitated tuberculosis contact investigation as the m-health is only one component of the intervention.

- Suggest changing reference to ‘ingredients based’ to ‘components based’ or ‘micro-costing’ approach.

- The authors have not provided the time-horizon of the study in the abstract. The time horizon for the cost analysis will impact the results reported and therefore, should be reported here in the abstract also.

- Conclusion- the statement ‘the cost of mHealth-facilitated contact investigation for TB was high ‘is not fully substantiated. High in what context? Over a one- year time horizon? Most technologies would not be developed for just one year of use. Suggest editing this sentence to align with the sentiment noted in the discussion i.e, that while they have high upfront development costs, they become more efficient with time, scale/coverage and the prevalence of TB in the community.

Introduction (main body of paper)

- With more than 10 million newly reported cases and 1.6 million deaths in 2018, tuberculosis (TB) is the leading global infectious disease cause of mortality (1). This refers to cases in 2018. Is TB still the leading global infectious disease cause of mortality in 2021? Edit wording to reflect that it is still the case in 2021.

- CI should be in parenthesis.

- However, barriers to acceptance and completion, operational complexities and resource constraints have limited wide adoption of CI in resource-limited setting – back this up with references please.

- It is unclear whether the authors have sufficiently summarised all relevant background literature. What were the results of the efficacy trial? The reference Davis et al, 2019 concluded that Home-based, SMS-facilitated evaluation did not improve completion or yield of household TB contact investigation, likely due to challenges delivering the intervention components. This needs to be reported as it is relevant to this study.

- Also, what is the breadth of the literature in this space too? Are there any other comparable costing /cost-effectiveness studies?

As with many other interventions utilizing mobile health (mHealth) platforms, however, mHealth-facilitated CI for TB requires substantial up-front resource outlays to establish the necessary infrastructure (9,10) . Reference 10 - Htet KKK, Liabsuetrakul T, Thein S, evaluates an intervention appeared to not have an M-health component? so can this reference support this statement?

- The authors state we conducted a comprehensive assessment of the costs of development, implementation, and operation of mHealthfacilitated, patient-centered CI for TB in Kampala, Uganda. The rationale for a partial cost analysis (and not a full cost-effectiveness analysis) is not quite clear at this point.

Setting:

- Suggest using the word implementation instead of execution. Implementation is more widely accepted terminology.

Figure 1- information in Figure 1 is not readable. Please re- do this figure so that all content can be read.

Estimates of program development costs and program execution costs:

- These sections provide clear information of methods but do need to align closely with the headings in the supplementary tables and should at least explain the phases so that the reader understands this without needing to go to the supplementary tables. Perhaps set out in the phases?

- Be clear how much of the overhead building costs are being apportioned to this intervention- I imagine those costs are being apportioned to many other interventions/ programs in reality?

- As you are only costing this arm of the original trial, it is important that all costs are associated in some way to the m-health component, otherwise what is being described may be more reflective of the cost of a tuberculosis contact investigation program generally rather than the addition of m-health.

Supplementary table 1:

- Supplementary table 1 has many acronyms that are not described. Please add a key providing description of all acronyms.

Figure 2 needs to explain somewhere what the contact positivity rate is (i.e., what makes up this rate?)

Analysis:

- This section is clear.

Sensitivity analysis:

- Inclusion of extended time horizon is appropriate as m-health technology usually would continue to be useful for a number of years not just 1 year as per the trial so this is good to see.

- Testing coverage/scale is also important as m-health likely works on economies of scale to offset the initial investment or sunk costs. The prevalence of disease clearly impacts here as well.

- These SA’s are important for interpreting results.

Discussion section:

The discussion section does summarise key study findings and describes how they support conclusions however a couple of things:

- In relation to the statement ‘Specifically, we found that 76% of the total cost of the program was incurred during development, before the enrollment of a single participant’. This is often the case with investment in new technologies. It would be good to note this fact with reference to the broader literature.

- All other sections of the discussion are good including the section on limitations of the study which is comprehensive.

Conclusion marries up with the discussion.

References

Please check that all references are using the standard PLOS style. There seems to be some variation in style. Reference 1 for example has fully capitalised title and states world health organ. This needs editing.

6. PLOS authors have the option to publish the peer review history of their article (what does this mean?). If published, this will include your full peer review and any attached files.

Reviewer #1: **Yes: **Denny John

Reviewer #2: No

---

## [Author Response · Author response to Decision Letter 0]

8 Feb 2022

February 7, 2022

Dr Joerg Heber, Editor in Chief

PLOS ONE

Dear Dr. Heber,

Thank you for your review of our manuscript entitled “A cost analysis of implementing mobile-health facilitated tuberculosis contact investigation in a low-income setting.” We appreciate the careful review that the reviewers have provided, and we have responded in bold text below to each of their comments, with new text added to the manuscript quoted in orange text:

Responses to Overall Review from the Editor:

1. Provide a clear description of the program, and indicate which activities were done in the clinics, and which ones were part of the community initiative and why.

Thank you for this suggestion. We have described the program in more detail in the Materials and Methods section, under the Study Design and Setting subheading, in Paragraph 1 on Page 4.

“In this setting, TB contact investigation involved CHWs visiting the homes of TB patients, screening all contacts for TB symptoms, and recording their findings using a customized electronic survey application (CommCare, Dimagi, Boston, USA). The application employed decision-support logic to identify contacts requiring evaluation for TB and prompted CHWs to collect a sputum sample and offer HIV testing to eligible household members. The application also delivered personalized, automated text messages to each participant providing follow-up instructions, clinic visit reminders, and TB test results. In the routine care arm, automated text messages were not sent, and all contacts needing TB evaluation were referred to the clinic. The home-based strategy sought to increase the proportion of contacts fully evaluated for TB by reducing the need for contacts to travel to clinics.”

2. The cost calculations are said to be done from the health system perspective yet waiting times and travel costs for patients have been included. Please separate the health system costs and the patient costs.

Thank you for this comment. To clarify, we did not collect any data on patient costs for this study. Waiting times actually refer to the time spent by CHWs unoccupied while waiting to enroll new patients at the clinic. Similarly, travel costs refer to the amount of money that CHWs spent on travel to visit households. A full description of how these costs apply to the health system is provided on Pages 4 and 5 of the Materials and Methods Section, under sub-heading, Estimation of Program Implementation Costs.

3. Capital costs for the overall TB treatment have also been included, however this was an add on program, TB services costs were already incurred, it would be more important to the policy makers to understand the additional costs that will be required to include the mHealth component.

Thank you for this comment. To clarify, we did not include capital costs for the overall TB treatment program in these estimates, but only the incremental costs. As shown in Table 2a, capital costs included software purchases, information technology (IT) hardware for the mHealth intervention, training in the use of the mHealth component, a vehicle for overall site supervision, and activities to adapt the intervention that our Ugandan partners felt would be necessary for programmatic implementation in different local settings.

4. There is no information on the messages that were sent and the associated costs of sending the messages.

Thank you for this question. We originally aggregated messaging costs with capital costs for IT hardware, but we now recognize that these are better classified as recurrent costs. We have added these costs (i.e., of the 10,000 text messages costing $50 in total that were purchased to cover 5 years of the program) as an independent cost component in Table 2a and Table 3. We have also updated the capital costs in Table 2a, Table 2b, and Table 3, and in the relevant narratives in the Results sections (see Page 11, Paragraph 2). Given the low cost of messaging, we observed no change in the overall distribution of macro- or micro-cost estimates.

5. It appears that the research related costs such as patient enrolment and program evaluation have been included in the costing, as indicated above, policy makers would be more interested in the additional costs of implementing the program.

Thank you for this comment. We agree that only programmatic costs should be included. Our assumption in including these costs was that patient enrollment includes standard contact investigation activities such as counselling, inviting participation, and obtaining standard clinical consent for the household visit. We also assume that program evaluation is a standard part of implementation and quality assurance, and consequently the estimated costs of program evaluation are small, consistent with this purpose. 

To clarify our approach in the manuscript, we have now replaced “enrollment” with “recruitment” throughout. The nature of patient recruitment activities is described in Supplemental Table 2. Finally, we have added to our description of program evaluation the qualifying phrase “for routine quality assurance” in Paragraph 1 on Page 6 in the Materials and Methods section, under the subheading Estimation of Program Implementation Costs.

Revisions made to meet journal requirements:

We have revised the manuscript in accordance with the style templates.

We have corrected the grant numbers in both locations.

3. In your Data Availability statement, you have not specified where the minimal data set underlying the results described in your manuscript can be found. PLOS defines a study's minimal data set as the underlying data used to reach the conclusions drawn in the manuscript and any additional data required to replicate the reported study findings in their entirety. All PLOS journals require that the minimal data set be made fully available. For more information about our data policy, please see http://journals.plos.org/plosone/s/data-availability.Upon re-submitting your revised manuscript, please upload your study’s minimal underlying data set as either Supporting Information files or to a stable, public repository and include the relevant URLs, DOIs, or accession numbers within your revised cover letter. For a list of acceptable repositories, please see http://journals.plos.org/plosone/s/data-availability#loc-recommended-repositories. Any potentially identifying patient information must be fully anonymized. Important: If there are ethical or legal restrictions to sharing your data publicly, please explain these restrictions in detail. Please see our guidelines for more information on what we consider unacceptable restrictions to publicly sharing data: http://journals.plos.org/plosone/s/data-availability#loc-unacceptable-data-access-restrictions. Note that it is not acceptable for the authors to be the sole named individuals responsible for ensuring data access. We will update your Data Availability statement to reflect the information you provide in your cover letter.

Thank you for the guidance. All data used in this analysis was de-identified and have been made available through the Dryad public repository. 

Yes, of course. We have added a full ethics statement at the end of the Materials and Methods section, in the second -to-last paragraph on Page 7.

“Ethical Considerations

The School of Medicine Research Ethics Committee at Makerere University; the Uganda National Council for Science and Technology; and the Yale University Human Investigation Committee approved the study protocol, informed consent and assent forms.”

5. Please upload a new copy of Figure 2 as the detail is not clear. Please follow the link for more information: https://blogs.plos.org/plos/2019/06/looking-good-tips-for-creating-your-plos-figures-graphics/" https://blogs.plos.org/plos/2019/06/looking-good-tips-for-creating-your-plos-figures-graphics/.

Thank you for these suggestions. We have prepared a new figure in the recommended format.

Reviewers' comments:

Reviewer's Responses to Questions

Comments to the Author

1. Is the manuscript technically sound, and do the data support the conclusions?

Reviewer #1: Yes

Reviewer #2: Yes

2. Has the statistical analysis been performed appropriately and rigorously?

Reviewer #1: Yes

Reviewer #2: Yes

3. Have the authors made all data underlying the findings in their manuscript fully available?

The PLOS Data policy requires authors to make all data underlying the findings described in their manuscript fully available without restriction, with rare exception (please refer to the Data Availability Statement in the manuscript PDF file). The data should be provided as part of the manuscript or its supporting information, or deposited to a public repository. For example, in addition to summary statistics, the data points behind means, medians and variance measures should be available. If there are restrictions on publicly sharing data—e.g., participant privacy or use of data from a third party—those must be specified.

Reviewer #1: Yes

Reviewer #2: Yes

4. Is the manuscript presented in an intelligible fashion and written in standard English?

Reviewer #1: Yes

Reviewer #2: No

We have now edited the manuscript to improve its clarity and readability.

5. Review Comments to the Author

Reviewer #1: The manuscript is well-written and methodologically sound. For further improvement the authors can check the manuscript against the transparency standards for cost estimates from the reference: Fukuda H, Imanaka Y: Assessment of transparency of cost estimates in economic evaluations of patient safety programmes. J Eval Clin Pract. 2009, 15: 451-459. 10.1111/j.1365-2753.2008.01033.x.

Thank you for sharing this reference. We have verified our adherence to these standards and stated this in the “Discussion” section at the bottom of Page 9.

“Our analysis adheres to transparency recommendations on scope and accuracy, and captures unit cost estimates of all the key program components.(26) While we do not report granular expenditures (e.g., number of sputum cups purchased), all macro cost calculations were derived from itemized unit costs and quantities. Furthermore, our method of micro cost estimation also provides the recommended highest level of accuracy, estimating the cost of each discrete implementation activity based on actual consumption.”

Reviewer #2: Specific feedback to the Author:

Abstract

Introduction: Mobile health (mHealth) applications may improve timely access to health services and improve patient-provider communication but may be costly to implement, especially in resource-limited settings. This statement does not quite reflect the rationale for the study. Suggest changing to reflect that development/ upfront investment in m- interventions maybe resource intensive.

We agree, and we have revised the Introduction section of the Abstract on Page 2) as follows:

“Mobile health (mHealth) applications may improve timely access to health services and improve patient-provider communication, but the upfront costs on its implementation may be prohibitive, especially in resource-limited settings.”

- The authors need to provide a clear aim/ research question in their abstract that aligns with the title and accurately represents the intervention. In the abstract, the authors state: We assessed the cost of development and execution of an mHealth-based program for contact investigation of tuberculosis (TB) in Kampala, Uganda however, this program has a m-health component alongside face to face intervention (i.e., community Health Worker (CHW)-led home-based specimen collection) therefore the authors should continue to refer to this program as a mobile-health facilitated tuberculosis contact investigation as the m-health is only one component of the intervention.

Thank you for this suggestion. We now describe the implementation strategy as the “mHealth-facilitated TB contact investigation” throughout the manuscript.

- Suggest changing reference to ‘ingredients based’ to ‘components based’ or ‘micro-costing’ approach.

Thank you for this suggestion. We now describe the bottom-up costing as “components-based.”

- The authors have not provided the time-horizon of the study in the abstract. The time horizon for the cost analysis will impact the results reported and therefore, should be reported here in the abstract also.

Thank you for this suggestion. Time horizons have now been integrated into the abstract and elsewhere in the manuscript. Figure 1 also provides a conceptual illustration of the costing timeline. 

“We estimated total costs per contact investigated and per TB-positive contact identified in 2018 US dollars, one and five years after program implementation.”

- Conclusion- the statement ‘the cost of mHealth-facilitated contact investigation for TB was high ‘is not fully substantiated. High in what context? Over a one- year time horizon? Most technologies would not be developed for just one year of use. Suggest editing this sentence to align with the sentiment noted in the discussion, i.e., that while they have high upfront development costs, they become more efficient with time, scale/coverage and the prevalence of TB in the community.

Thank you again for these observations and helpful suggestions. We have revised the Abstract’s Conclusion on Page 2 to emphasize that high upfront development costs may become more manageable if the intervention can be implemented at scale and over time. 

“Over 75% of all costs of the mHealth-facilitated TB contact investigation strategy were dedicated to establishing mHealth infrastructure and capacity. Implementing the mHealth strategy at scale and maintaining it over a longer time horizon could help decrease development costs as a proportion of total costs.”

Introduction (main body of paper)

- With more than 10 million newly reported cases and 1.6 million deaths in 2018, tuberculosis (TB) is the leading global infectious disease cause of mortality (1). This refers to cases in 2018. Is TB still the leading global infectious disease cause of mortality in 2021? Edit wording to reflect that it is still the case in 2021.

Thank you for this comment. Because COVID is now the leading infectious cause of death, we have revised the language as follows:

“Tuberculosis (TB) is among the leading causes of mortality due to an infectious disease worldwide with approximately 7 million new cases of TB diagnosed in 2020.”

- CI should be in parenthesis.

We have put CI in parentheses upon first reference in the abstract and in the main text.

- However, barriers to acceptance and completion, operational complexities and resource constraints have limited wide adoption of CI in resource-limited setting – back this up with references please.

Thank you, references supporting this statement have now been added:

2. Armstrong-Hough M, Turimumahoro P, Meyer AJ, Ochom E, Babirye D, Ayakaka I, et al. Drop-out from the tuberculosis contact investigation cascade in a routine public health setting in urban Uganda: A prospective, multi-center study. PLoS One. 2017;12(11).

8. Ayakaka I, Ackerman S, Ggita JM, Kajubi P, Dowdy D, Haberer JE, et al. Identifying barriers to and facilitators of tuberculosis contact investigation in Kampala, Uganda: A behavioral approach. Implement Sci. 2017.

9. Ngamvithayapong-Yanai J, Luangjina S, Thawthong S, Bupachat S, Imsangaun W. Stigma against tuberculosis may hinder non-household contact investigation: a qualitative study in Thailand. Public Health Action. 2019.

- It is unclear whether the authors have sufficiently summarized all relevant background literature. What were the results of the efficacy trial? This needs to be reported as it is relevant to this study.

Yes, this is important. In addition to the above noted references, we have also added the results of the parent trial in the Study Design and Setting subsection of the Materials and Methods section in Paragraph 1 on Page 5 and cited the publication that reported the original results. 

“The overall trial observed a marginal probability of completing TB evaluation of 14% (95% CI 8-20 14%) in intervention households and 15% (95% CI 9-21) in routine care households, representing a difference of -1% (95% CI -9% to 7%, p=0.81.”

- Also, what is the breadth of the literature in this space too? Are there any other comparable costing /cost-effectiveness studies?

We have now referenced and discussed the following literature that references the cost of mHealth programs in low- and middle-income settings in the Discussion section, in Paragraph 3, on Page 14.

21. Prinja S, Bahuguna P, Gupta A, Nimesh R, Gupta M, Thakur JS. Cost effectiveness of mHealth intervention by community health workers for reducing maternal and newborn mortality in rural Uttar Pradesh, India. Cost Eff Resour Alloc. 2018

22. Chang LW, Kagaayi J, Nakigozi G, Serwada D, Quinn TC, Gray RH, et al. Cost analyses of peer health worker and mHealth support interventions for improving AIDS care in Rakai, Uganda. AIDS Care - Psychol Socio-Medical Asp AIDS/HIV. 2013.

25. Nsengiyumva NP, Mappin-Kasirer B, Oxlade O, Bastos M, Trajman A, Falzon D, et al. Evaluating the potential costs and impact of digital health technologies for tuberculosis treatment support. Eur Respir J. 2018.

As with many other interventions utilizing mobile health (mHealth) platforms, however, mHealth-facilitated CI for TB requires substantial up-front resource outlays to establish the necessary infrastructure (9,10). Reference 10 - Htet KKK, Liabsuetrakul T, Thein S, evaluates an intervention appeared to not have an M-health component? so can this reference support this statement? for this question. 

We have replaced the Htet KKK et al reference and reworded the literature we found on cost evaluations in the text below, found in the last paragraph on Page 3. 

“The strategy was feasible and acceptable but not more effective because of implementation challenges (11,12). Nonetheless, another challenge in mHealth field is the limited and heterogeneous evidence on the costs and cost effectiveness of mHealth strategies (13), including some evidence of high up-front costs (14), which may act as a barrier to ongoing research and innovation.”

13. Iribarren SJ, Cato K, Falzon L, Stone PW. What is the economic evidence for mHealth? A systematic review of economic evaluations of mHealth solutions. PLoS One. 2017.

14. Larsen-Cooper E, Bancroft E, Rajagopal S, O’Toole M, Levin A. Scale matters: A cost-outcome analysis of an m-health intervention in Malawi. Telemed e-Health. 2016.

- The authors state we conducted a comprehensive assessment of the costs of development, implementation, and operation of mHealth facilitated, patient-centered CI for TB in Kampala, Uganda. The rationale for a partial cost analysis (and not a full cost-effectiveness analysis) is not quite clear at this point.

We had originally proposed a cost effectiveness analysis, but this was not possible because the mHealth intervention strategy was not more effective than routine contact investigation for TB. Therefore, we are reporting a cost analysis only, to provide real world estimates of the expenditure of implementing such programs. We have clarified this in the final paragraph of the Introduction on Page 3 (partially cited above, provided again here for full context):

“To address these challenges, we developed a home-based, mHealth-facilitated household CI strategy and evaluated it in a pragmatic, prospective, household randomized trial (10). Compared to routine CI delivered by community health workers (CHWs), the mHealth-facilitated CI intervention included home-based HIV testing and TB evaluation, collection and transport of sputum samples, and follow-up communications using automated short messaging services (SMS). The strategy was feasible and acceptable but not more effective than routine contact investigation because of implementation challenges (11, 12). Nonetheless, another area of uncertainty in the mHealth field is the limited and heterogeneous evidence on the costs and cost effectiveness of mHealth strategies (13), including some evidence of high up-front costs (14), which may in turn act as a barrier to ongoing research and innovation. Therefore, to characterize the resource implications of such health interventions more fully, we conducted a comprehensive assessment of the costs of development, implementation, and maintenance of home-based, mHealth-facilitated TB CI in Kampala, Uganda.”

Setting:

- Suggest using the word implementation instead of execution. Implementation is more widely accepted terminology.

Thank you for this comment; we have revised accordingly throughout the manuscript.

Figure 1- information in Figure 1 is not readable. Please re- do this figure so that all content can be read.

We apologize for this. We have uploaded a revised figure that adheres to journal guidance.

Estimates of program development costs and program execution costs:

- These sections provide clear information of methods but do need to align closely with the headings in the supplementary tables and should at least explain the phases so that the reader understands this without needing to go to the supplementary tables. Perhaps set out in the phases?

Thank you. We have now provided additional information in the body of the Materials and Methods subsection entitled “Estimation of program development cost” on Page 5 to expand description of the supplementary tables in the main text described below:

“Cost components were appropriately mapped to specific thematic expenditure categories: human resource costs, capital costs, recurrent costs, overhead costs and building space costs. Human resource costs included a coordinator, data manager, laboratory manager and IT officer. Capital costs included investment in hardware and software for mHealth, a vehicle, and cost to adapt the intervention to the local setting. Recurrent costs included expenditure on consumables such as lab supplies, internet, and text messages. Overhead costs included operational costs such as those for supervision teams and patient care at the clinic. Building space was the space occupied by the supervision teams and patient rooms.”

- Be clear how much of the overhead building costs are being apportioned to this intervention- I imagine those costs are being apportioned to many other interventions/ programs in reality?

Thank you for this comment. We have added text to address this question at the bottom of Page 5 and the top of Page 6, in the Materials and Methods section under the sub-header “Estimation of program implementation costs”: 

“The cost of building space utilized for patient services was approximated as 5% of the cost of the entire building and operational costs for the program as 6.7% of clinic operational costs.”

- As you are only costing this arm of the original trial, it is important that all costs are associated in some way to the m-health component, otherwise what is being described may be more reflective of the cost of a tuberculosis contact investigation program generally rather than the addition of m-health.

Thank you for this comment. Only the study arm that was exposed to the mHealth components was considered for this analysis and we have only included costs associated with the mHealth components in some way. We have provided a general overview of our approach in the second paragraph of the Materials and Methods section on Page 4, and have edited this section to clarify that costing is focused only the mHealth components:

“To comprehensively evaluate the costs of mHealth-facilitated intervention, we divided the program into two phases and evaluated the costs accrued in each phase.”

We further elaborate on this process in the “Estimation of program development costs” and “Estimation of program implementation costs” that immediately follow, on Pages 4 and 5. Finally, Supplementary Table 1 summarized all included costs.

Supplementary table 1:

- Supplementary table 1 has many acronyms that are not described. Please add a key providing a description of all acronyms.

Yes, of course, we have now defined all abbreviations.

Figure 2 needs to explain somewhere what the contact positivity rate is (i.e., what makes up this rate?)

Thank you. We have now revised the Figure 2 Legend on Page 13 to explain that the contact positivity rate is obtained by dividing the number of contacts diagnosed with TB by the number of contacts evaluated per year.

“...the average annual contact positivity rate (number of contacts diagnosed with TB divided by the number of contacts evaluated)”

Analysis:

- This section is clear.

Sensitivity analysis:

- Inclusion of extended time horizon is appropriate as m-health technology usually would continue to be useful for a number of years not just 1 year as per the trial, so this is good to see.

Thank you.

- Testing coverage/scale is also important as m-health likely works on economies of scale to offset the initial investment or sunk costs. The prevalence of disease clearly impacts here as well.

Yes, we entirely agree and have emphasized this in the Abstract Conclusion on Page 2 and elsewhere in the manuscript:

“Over 75% of all costs of the mHealth-facilitated TB contact investigation strategy were dedicated to establishing mHealth infrastructure and capacity. Implementing the mHealth strategy at scale and maintaining it over a longer time horizon could help decrease development costs as a proportion of total costs.”

- These SA’s are important for interpreting results.

Thank you for these comments.

Discussion section:

The discussion section does summarize key study findings and describes how they support conclusions however a couple of things:

- In relation to the statement ‘Specifically, we found that 76% of the total cost of the program was incurred during development, before the enrollment of a single participant’. This is often the case with investment in new technologies. 

It would be good to note this fact with reference to the broader literature.

Thank you. In the subsequent sentence in the first paragraph of the Discussion at the bottom of Page 13, we talk about how our findings compare to other mHealth programs. 

“Specifically, we found that 76% of the total cost of the program was incurred during development, before the recruitment of a single participant. The total program cost was therefore estimated at $320-$348 per contact investigated and $8,873-$9,652 per new diagnosis of TB – a high cost relative to many other programs(13,21,22). Importantly, these costs could be reduced to under $600 per new TB diagnosis simply by expanding capacity (extending from one to five years, increasing the number of clinics participating, and optimizing the volume of household contacts evaluated in each clinic). These findings illustrate that, when implementing mHealth and other interventions with substantial development costs in resource-limited settings, the feasible scope and duration of the program, as well as the expected yield, must be considered to evaluate whether a meaningful return on investment is likely.”

We further expand on this statement in Paragraph 3 on Page 14 of the Discussion section.

“More broadly, the cost implications of digital health interventions in low- and middle-income countries vary depending on their scope and purpose. For example, a digital adherence program in Brazil applying two way SMS cited a cost of $65 ($53–$105) per person enrolled (25), while an mHealth support intervention for HIV in Uganda cited an annual cost of $2.35 per patient enrolled.(22) A study in India on an mHealth-facilitated intervention to improve CHW counseling skills in maternal and newborn health suggested that start-up costs represented only 9% of total costs; this program was able to operate at a cost of $20 per woman registered.(21) Taken as a whole, these findings illustrate the heterogeneous economic implications of mHealth-facilitated health interventions designed for resource-limited settings, as well as the importance of considering feasible program scope and costs of program development before embarking on large investments in mHealth infrastructure and capacity.”

- All other sections of the discussion are good including the section on limitations of the study which is comprehensive.

Conclusion marries up with the discussion.

Thank you for these comments.

References

Please check that all references are using the standard PLOS style. There seems to be some variation in style. Reference 1 for example has fully capitalized title and states world health organ. This needs editing.

Thank you, all references have been added in similar font and the referencing style is Vancouver.

6. PLOS authors have the option to publish the peer review history of their article (what does this mean?). If published, this will include your full peer review and any attached files.

Do you want your identity to be public for this peer review? For information about this choice, including consent withdrawal, please see our Privacy Policy.

Reviewer #1: Yes: Denny John

Reviewer #2: No

Sincerely,

Patricia Turimumahoro, MBChB, MPH

J. Lucian Davis, MD, MAS

---

## [Editor Report · Decision Letter 1]

23 Feb 2022

A cost analysis of implementing mobile-health facilitated tuberculosis contact investigation in a low-income setting

PONE-D-21-01299R1

Dear Dr. J. Lucian Davis,

We’re pleased to inform you that your manuscript has been judged scientifically suitable for publication and will be formally accepted for publication once it meets all outstanding technical requirements.

Kind regards,

Limakatso Lebina, MBChB, Ph.D.

Academic Editor

PLOS ONE

Additional Editor Comments (optional):

You have been able to address all comments in the revised manauscript.
---

## [Editor Report · Acceptance letter]

15 Mar 2022

PONE-D-21-01299R1 

A cost analysis of implementing mobile health facilitated tuberculosis contact investigation in a low-income setting 

Dear Dr. Davis:

I'm pleased to inform you that your manuscript has been deemed suitable for publication in PLOS ONE. Congratulations! Your manuscript is now with our production department. 

Kind regards, 

on behalf of

Dr. Limakatso Lebina 

Academic Editor

PLOS ONE